

# The antioxidant effects of butylated hydroxytoluene on cryopreserved goat sperm from a proteomic perspective

Chunyan Li[1,2,3], Larbi Allai[1,4], Jiachong Liang[1,2,3], Chunrong Lv[1,2,3], Xiaoqi Zhao[1,3], Xiaojun Ni[1,3], Guoquan Wu[1,2,3], Weidong Deng[5], Bouabid Badaoui[6] and Guobo Quan[1,2,3]

[1] Yunnan Animal Science and Veterinary Institute, Kunming, China
[2] Yunnan Provincial Genebank of Livestock and Poultry Genetic Resources, Kunming, China
[3] Yunnan Provincial Engineering Research Center of Animal Genetic Resource Conservation and Germplasm Enhancement, Kunming, China
[4] Higher School of Technology Sidi Bennour, Chouaib Doukkali University, El Jadida, Morocco
[5] School of Animal Science and Technology, Yunnan Agricultural University, Kunming, China
[6] Faculty of Sciences, Mohammed V University, Rabat, Morocco

Corresponding author
Guobo Quan,
waltq20020109@163.com

## ABSTRACT

At present, there are few reports about the proteomics changes provoked by butylated hydroxytoluene (BHT) supplementation on cryopreserved semen in mammals. Thus, we aimed to evaluate the effects of different concentrations of BHT on goat sperm and to investigate the proteomics changes of adding BHT to cryopreserved goat (*Capra hircus*) sperm. Firstly, semen samples were collected from four goats, and frozen in the basic extenders containing different concentrations of BHT (0.5 mM, 1.0 mM, 2.0 mM) and a control without BHT, respectively. After thawing, the protective effects of dose-dependent replenished BHT to the freezing medium on post-thaw sperm motility, integrities of plasma membrane and acrosome, reactive oxygen species levels were confirmed, with 0.5 mM BHT being the best (B group) as compared to the control (without BHT, C group). Afterwards, TMT-based quantitative proteomic technique was performed to profile proteome of the goat sperm between C group and B group. Parallel reaction monitoring was used to confirm reliability of the data. Overall, 2,476 proteins were identified and quantified *via* this approach. Comparing the C and B groups directly (C *vs.* B), there were 17 differentially abundant proteins (DAPs) potentially associated with sperm characteristics and functions were identified, wherein three were upregulated and 14 were downregulated, respectively. GO annotation analysis demonstrated the potential involvement of the identified DAPs in metabolic process, multi-organism process, reproduction, reproductive process, and cellular process. KEGG enrichment analysis further indicated their potential roles in renin-angiotensin system and glutathione metabolism pathways. Together, this novel study clearly shows that BHT can effectively improve quality parameters and fertility potential of post-thawed goat sperm at the optimal concentration, and its cryoprotection may be realized through regulation of sperm metabolism and antioxidative capability from the perspective of sperm proteomic modification.

## INTRODUCTION

Semen cryopreservation is an important technology, which can preserve and utilize sperm for a long time, facilitating rapid dissemination of excellent genetic material around the world. Moreover, semen cryopreservation promotes the development and application of artificial insemination technology (AI), (*Leboeuf, Restall & Salamon, 2000*). However, the cryopreservation process has a deleterious impact on sperm normal physiology, reducing the quality because of adverse factors such as ultra-low temperature stress, osmotic stress and oxidative stress, ultimately, reducing the longevity and ability to fertilize such as vitality, acrosome reaction rate and egg fertilization rate (*Peris-Frau et al., 2020*).

Among these injuries caused by the cryopreservation process, oxidative lesion has been shown to greatly threaten sperm structures such as acrosome, mitochondria, nucleus DNA, and the functions such as motility and its ability to fusion with oocyte (*Pariz et al., 2019*). Given this, numerous antioxidants, previously reported, have been tested and added into freezing extenders to resist oxidative stress and improve fertility of post-thaw sperm (*Dutta, Majzoub & Agarwal, 2019*). In general, these antioxidants have been divided into two types: (1) enzymatic antioxidants, such as glutathione catalase (GSH-Px), glutathione reduction enzyme (GR), superoxide dismutase (SOD), glutathione-S-transferase (GST), melatonin, *etc.*, (2) non-enzymatic antioxidants, such as glutathione (GSH), urate, vitamin, carbohydrate, *etc* (*Amidi et al., 2016*; *Jia et al., 2021*; *Riesco et al., 2021*; *Turk et al., 2022*; *Li et al., 2023*). Currently, many studies have been performed to assess the cryoprotective effects of these antioxidants on sperm during the cryopreservation process (*Pezo et al., 2021*; *Ribas-Maynou et al., 2021*; *Tiwari et al., 2022*).

Thereinto, butylated hydroxytoluene (BHT) is being investigated as a component to cryomedia in different animal species, including goat (*Memon et al., 2011*), human (*Merino et al., 2015*), boar (*Sus scrofa*) (*Trzcinska et al., 2015*), cat (*Jara et al., 2019*), and buffalo (*Nain et al., 2022*), because of its safety and notable anti-oxidative potential. BHT is a synthetic analogue of vitamin E, characterized with the low polarity and high lipid solubility (*Yehye et al., 2015*; *Achar et al., 2020*). It can be used to check auto-oxidation reaction of lipid bilayer and membrane by converting peroxy radicals into hydroperoxides, thus inhibiting lipid peroxidation (LPO). Besides, BHT also scavenges reactive oxygen species (ROS) from the surroundings of sperm, thereby minimizing the cold shock and increasing antioxidant defense of sperm during the cryopreservation process (*Merino et al., 2015*; *Khumran et al., 2019*). Although several studies have showed beneficial effects of BHT on sperm quality after cryopreservation, the specific cryoprotective mechanisms underlying have not been fully elucidated.

It is known that sperm is a specialized cell with inactive transcription, and therefore, once sperm has left the male reproductive tract, they rely mainly on the static population action of proteins and metabolites to maintain their function, prior to fertilization with oocytes, especially, the principal roles of proteins cannot be neglected (*Hermo et al., 2010*; *Chauvin et al., 2012*). Based on these facts that antioxidants play important roles in enhancement of frozen-thawed sperm antioxidative capability and the action mechanisms of most antioxidants remain unclear, taking BHT as an example, the aim of this study

endeavors to: (1) ascertain the optimal BHT concentration conducive to the preservation of post-thaw goat sperm quality, (2) elucidate the potential cryoprotective mechanism of this optimal BHT mainly from the perspective of goat sperm proteome. The identification of differential proteins coupled with their correlative bioinformatic analysis will deepen our understanding on the molecular mechanisms of BHT on mammalian sperm.

## MATERIAL AND METHODS

### Study design and workflow
The objective was to evaluate cryoprotective effects of different concentrations of BHT (0.0 mM, 0.5 mM, 1.0 mM, and 2.0 mM) on goat sperm during the cryopreservation process, indices related to sperm quality such as the motility, ROS levels, plasma membrane and acrosome integrities were assessed in current study. Thus, the optimal effect concentration of BHT was going to be determined. Afterwards, TMT-based quantitative proteomic technique was used to investigate potential effects of BHT on proteome of cryopreserved goat sperm. Overall workflow was shown in Fig. S1.

### Semen preparation
During the whole experiment, the authors strictly complied with Regulations on the Administration of Laboratory Animals (Order-No.2 of the State Science and Technology Commission of the People's Republic of China, 1988) and Regulations on the Administration of Experimental Animals of Yunnan Province (the Standing Committee of Yunnan Provincial People's Congress 2007.10). All semen samples were acquired from Yi Xingheng Animal Science and Technology Co. Ltd (Kunming, Yunnan Province, China). Four males of Yunshang black goats with similar ages (2~3 years old) were selected, and kept in the same condition of feeding and management. Fresh semen (two ejaculates per male in 10 min) were collected *via* an artificial vagina method. Then, semen from each male was pooled and kept at 37 °C, and the quality was immediately assessed using a computer-assisted sperm analysis system (CASA, Microptic, Barcelona, Spain). Only semen meeting stringent quality criteria were accepted (volume of each semen $\geq$ 0.8 mL, sperm with concentration $\geq$ 2.5 $\times$ $10^9$/mL, and the motility $\geq$ 80%) (*Liu et al., 2019*). Pooled semen from each male was subsequently divided into four aliquots. One was used as sperm source of cryopreserved control group (C group, which was diluted using medium A), while the remaining three were used as sperm sources of cryopreserved experiment groups (B groups, which were diluted using medium B, respectively).

### Semen freezing medium
Above four semen aliquots of each male were lightly diluted with two types of frozen media (media A and B) at RT (25 ° C), until a final sperm concentration reached about $3 \times 10^8$ /mL, respectively (*Succu et al., 2011*). Medium A was made up of 254 mM Tris, 85 mM citric acid, 70 mM fructose, 1% (w/v) soybean lecithin, 6% (v/v) glycerin, $1 \times 10^4$ IU penicillin and $1 \times 10^4$ IU streptomycin. Medium B consisted of three subgroups being made up the medium A and three concentrations of butylated hydroxytoluene (BHT, Sigma-Aldrich, USA) with 0.5, 1.0 and 2.0 mM, respectively (*Merino et al., 2015*).

## Freezing and thawing process

Extended semen samples were loaded into the labeled 0.25 mL straws (IMV technologies, L'Aigle, France), following by equilibration at 5 °C for 3 h. After that, freezing through being placed in liquid nitrogen vapor for 7 min (−80 °C), and directly into liquid nitrogen immediately for at least one week (*Ruiz-Diaz et al., 2020*). For the thawing process, frozen semen was thawed in water bath at 37 °C for 30 s (*Fang et al., 2019*; *Li et al., 2023*).

## Evaluation of sperm-quality associated indices

Motilities of all post-thaw sperm from the C and B groups were assessed *via* the CASA system (*Amann & Waberski, 2014*). Plasma membrane integrity was assessed by the hypo-osmotic swelling test (HOST) with the GENMED kit (GMS14017, GENMED Scientifcs Inc., Wilmington, MA, USA). Briefly, 10 µL of sperm sample was mixed with 100 µL of pre-heated hypo-osmotic solution, incubated at 37 °C for 30 min. 10 µL of the mixture was then smeared onto a pre-heated glass slide with a coverslip. Over 200 sperm were counted under a phase-contrast microscope (Axio Vert A1, Germany). Sperm with a coiled tail indicated its has intact plasma membrane. Integrity rate of sperm was calculated (*Zou et al., 2021*; *Li et al., 2023*). Additionally, acrosome integrity was evaluated by the fluorescein isothiocyanate labeled pea agglutinin (FITC-PSA) test, and the detailed method has been described in our previous report (*Jia et al., 2022*). For analysis of intracellular ROS production, 5 µL of post-thaw semen was diluted with 500 µL TALP solution, followed by 0.5 µL DCFH-DA and 5 µL PI for 60 min at 25 °C in darkness, the mixture was evaluated by a flow cytometer (BD Accuri™ C6, Franklin Lakes, NJ, USA) (*Najafi et al., 2018*).

## TMT-based proteomic investigation
### Sperm sample preparation

Post-thawing semen were centrifuged at 3,000× g for 15 min at 4 °C (JIDI-20RS, China). Collected sperm pellets were washed three times in 1× phosphate-buffered solution (1× PBS; Gibco, Thermo Scientific, Waltham, MS, USA) by centrifugation (3,000× g, 5 min, 4 °C), the pellet samples were immediately stored at −80 °C (*Zhu et al., 2020*).

### Protein extraction and digestion

For protein extraction, every 30 mg sperm pellet was resuspended in the lysis buffer containing 8 M urea (Sigma, St. Louis, MO, USA), 2 mM EDTA (Sigma, St. Louis, MO, USA), 10 mM dithiothreitol (Sigma, St. Louis, MO, USA) and 1% protease inhibitor cocktail (SparkJade, Shandong, China), and incubated for 2 min. Subsequently, the samples were sonicated three times on ice using a high intensity ultrasonic processor (Scientz-1200E, Hangzhou, China). The residual impurity was removed by centrifugation at 16,000× g for 10 min at 4 °C. Finally, proteins were precipitated with cold 15% trichloroacetic acid for 2 h at −20 °C. After centrifugation for 10 min at 4 °C, and the precipitate was further washed with cold acetone thrice. Protein concentration was quantified using a BCA protein assay kit (Beyotime, Shanghai, China, Table S1). The 15 µg extracted protein was individually separated by 10% SDS-PAGE, and the gel protein profile from each sample was presented in Fig. S2 (*Guo et al., 2019*).

A total of 300 μg proteins per sample was used for trypsin digestion. Briefly, each sample was added DTT to 100 mM and incubated for 5 min at 100 °C, followed by cooling and processing with 200 μL UA buffer containing 8M urea and 150 mM TrisHCl (Sigma, Burlington, MA, USA). The supernatant was discarded after centrifugation at 12,000×g for 15 min. The pellet was further processed with 200 μL UA buffer again. After centrifuging, 50 mM iodoacetamide (IAA, Sigma, Burlington, MA, USA) was added to alkylate the solution for 30 min at RT in darkness, followed by centrifugation at 12,000× g for 10 min. 100 μL UA buffer was added and centrifuged again. This step was followed by a buffer exchange with 100 μL of $NH_4HCO_3$ buffer and further centrifugation at 14,000× g for 10 min. The digestion process involved incubation with 60 μL of trypsin buffer (6 μg trypsin in 40 μL $NH_4HCO_3$ buffer) for 16–18 h at 37 °C.

### TMT labeling and fractionation of peptides

After trypsin digestion, peptides were initially desalted by Strata X C18 SPE column (Phenomenex, Signa, Innsbruck, Austria), and subsequently dried by vacuum centrifugation. The peptides (100 μg) from each sample were added to 0.5 M TEAB solution and processed using the 10-plex TMT kit (Thermo Fisher Scientific, Waltham, MA, USA). Briefly, the above peptides dissolved solution was incubated with the TMT regent (1 unit of labeling reagent was used for 100 μg of peptide), which was reconstituted in 24 μL anhydrous acetonitrile (CAN) for 2 h at RT. Then, four pooled fractions of C group have been labeled with 126 (C1), 127N (C2), 127C (C3) and 128N (C4) tags, while four pooled fractions of B group have been labeled with 128C (B1), 129N (B2), 129C (B3) and 130N (B4) tags, respectively. The reaction was stopped with 8% ammonium hydroxide. Differently labeled peptides were mixed equally, desalted and vacuum dried (*Huang et al., 2017*; *Muraoka et al., 2019*), then fractionation into fraction using high pH reverse-phase column from Pierce™ high pH reversed-phase peptide fractionation kit (Thermo Fisher Scientific, Waltham, MA, USA) according to the manufacturer's protocol. The peptides finally were combined into 10 fractions and dried by vacuum centrifugation.

### LC-MS/MS analysis

The peptides were loaded onto a Trap Column (100 μm ×20 mm, 5 μm, C18, Dr. Maisch GmbH) using 0.1% (v/v) solvent A (formic acid, FA). Separation was then performed on a chromatographic column (75 μm ×150mm, 3 μm, C18, Dr. Maisch GmbH) using an increase of solvent B (0.1% FA in 95% acetonitrile solution). The increasing profile was setting as follows: an increase from 2% to 8% for 2 min, 8% to 28% for 69 min, 28% to 40% for 8 min, then a rise to 100% for 2 min, and sustained at 100% for an additional 9 min, overall process was conducted at a constant flow rate of 300 nL/min on an EASY-nLC 1200 UPLC system (Thermo Fisher Scientific, Waltham, MA, USA). Then, mass spectrometry (MS) analysis was performed by a Q Exactive HF-X mass spectrometer (Thermo Fisher Scientific, Waltham, MA, USA) in the positive ion model and data-dependent acquisition for 90 min. A full scan range of MS was set to 350–1,800 m/z at a resolution of 60,000@m/z 200. Automatic gain control (AGC) target was set at 3E6 ions and the maximum injection time was set at 50 ms. Afterwards, MS/MS was performed in the same order. The setting parameters as following: resolution of MS2 scan was 45,000@m/z 200, AGC target was 1E5,

the maximum injection time was 50 ms, activation type was higher energy dissociation (HCD), isolation window was 1.2 m/z, and normalized collision energy was set as 32.

### Identification and quantification of proteins

A Proteome Discoverer 2.4 software (v.1.6.0.16) was used to retrieve and analyze the LC-MS/MS raw data. Specified search parameters were as follows: (1) a target-reverse database derived from https://www.uniprot.org/taxonomy/9925 protein database: Uniprot_Capra hircus (goat)_9925 (35,503 sequences).fasta; (2) quantitation type: TMT 10-plex isobaric labels; (3) mass tolerance: precursor ions tolerance: 10 ppm, and mass error tolerance of fragment ions: 0.02 Da; (4) digestion: trypsin/P; (5) modifications: fixed modifications: Carbamidomethyl (C), TMT6plex (K), TMT6plex (peptide N-term), and variable modifications: oxidation (M), acety (protein N-term); (6) FDR setting: $\leq$ 1%; (7) unique peptides per protein setting: $\geq$ 1.

### Bioinformatic analysis

Bioinformatics data were carried out using Perseus software (v.1.6.5.0), microsoft excel (v.2010) and R statistical computing software (v.4.0.5). Differentially significant abundant proteins (DAPs) were screened base on a fold change (FC) cutoff of >1.20 or <0.833, and $P$-value < 0.05. Protein abundance data were grouped together via hierarchical clustering. Information of sequence annotation was extracted from UniProtKB/Swiss-Prot, Gene Ontology (GO) and Kyoto Encyclopedia of Genes and Genomes (KEGG). GO and KEGG enrichment analyses were carried out with the Fisher's exact test, and FDR correction for multiple testing was also performed. Annotation informations of GO function were obtained from the UniProt-GOA database (http://www.ebi.ac.uk/GOA/) combined with Inter-ProScan tool (https://www.ebi.ac.uk/interpro/), which including three categories: biological process (BP), cellular component (CC) and molecular function (MF). The KASS service (https://www.genome.jp/tools/kaas/) combined with DAVID tool (https://david.ncifcrf.gov/) were performed to annotate proteins' KEGG informations (*Guan et al., 2021*). The enrichment analysis of GO and KEGG pathway with $P$-value < 0.05 was considered statistically significant. WoLF PSORT service (https://www.genscript.com/wolf-psort.html) was used to predict protein subcellular localization. String database (http://www.string-db.org/) combined with cytoscape software (v.3.9.1) were used for constructing protein-protein interaction (PPI) networks.

## Parallel reaction monitoring quantification

Quantification verification of sperm samples was performed using parallel reaction monitoring (PRM) assay, protocols of sperm samples preparation, total protein extraction and trypsin digestion were the same as the TMT LC-MS/MS procedure. A total of 2 μg peptides of each sample were analyzed using an Easy-nLC 1200 UPLC system (Thermo Fisher Scientific, Waltham, MA, USA). Afterwards, MS/MS analysis was further performed in a Q Exactive HF-X mass spectrometer (Thermo Fisher Scientific, Waltham, MA, USA). Running parameters of the mass spectrometer were set as follows: (1) analysis duration was 60 min in positive ion detection mode. (2) precursor ion scan range of MS was 300–1,200 m/z at a resolution of 60,000@m/z 200. (3) AGC target was 3E6 ions, and the maximum

**Table 1  Effects of BHT on the motility and motile parameters of post-thaw goat sperm.**

| Groups | | TM (%) | PM (%) | VCL (μm/s) | VAP (μm/s) | VSL (μm/s) | ALH (μm) | BCF (Hz) |
|---|---|---|---|---|---|---|---|---|
| C | | 53.05 ± 3.84[a] | 29.22 ± 2.06[a] | 79.31 ± 4.32[a] | 55.75 ± 0.84[a] | 48.54 ± 2.94[a] | 2.77 ± 0.15[a] | 6.93 ± 0.52[a] |
| B (mM) | 0.5 | 61.68 ± 0.58[b] | 39.22 ± 2.98[b] | 82.50 ± 4.47[a] | 64.97 ± 1.15[b] | 53.79 ± 0.68[a] | 2.97 ± 0.15[a] | 7.64 ± 0.35[a] |
| | 1.0 | 54.14 ± 2.33[ab] | 33.91 ± 1.71[ab] | 83.04 ± 2.10[a] | 63.38 ± 3.95[ab] | 51.65 ± 3.29[a] | 2.71 ± 0.18[a] | 6.97 ± 0.31[a] |
| | 2.0 | 51.98 ± 3.02[a] | 33.02 ± 2.61[ab] | 82.25 ± 3.76[a] | 61.10 ± 3.76[ab] | 48.70 ± 2.80[a] | 2.72 ± 0.15[a] | 7.35 ± 0.43[a] |

Notes.
Within the same column, values with different lowercase superscripts differ significantly ($P < 0.05$).

injection time (MIT) was 50 ms. Followed by performing the same order during MS2, and the setting parameters as following: resolution of MS2 scan was 30,000@m/z 200, AGC target was 1E6, the MIT was 100 ms, activation type was HCD, isolation window was 1.6 m/z, and normalized collision energy was 28. Raw data of PRM-MS/MS were searched through the MaxQuant search engine, following processing using Skyline (v.4.1) (*Zhang et al., 2022*).

## Statistical analysis

IBM SPSS26.0 software (SPSS Inc., Chicago, IL, USA) was used to distinguish differential data. Based on equal variance assumption, check of variance homogeneity and the multiple comparison analysis by Duncan's test were conducted. Prism software (GraphPad, v.6.07) was used to present graphs as means ± standard error of the mean (S.E.M). A two-tailed Fisher's exact test with Bonferroni correction ($P$-value $< 0.05$) was employed to test the enrichment of DAPs. Functional enrichment with an adjusted $P$-value $< 0.05$ was considered significant.

## RESULTS

### Effects of BHT on sperm quality-associated parameters

Result of the motility and motile parameters in goat sperm were shown in Table 1. Sperm in samples frozen with 0.5 mM BHT showed significantly higher total motility ($61.68 \pm 0.58\%$) and progressive motility ($39.22 \pm 2.98\%$) than control samples ($53.05 \pm 3.84\%$ and $29.22 \pm 2.06\%$, respectively) ($P = 0.044$ and $P = 0.012$). Concurrently, sperm in samples frozen with 0.5 mM BHT showed significantly higher average velocity (VAP, μm/s) than the control ($P = 0.039$). However, other motile parameters of post-thaw sperm such as curvilinear velocity (VCL, μm/s), rectilinear velocity (VSL, μm/s), sway amplitude (ALH, μm) and whip frequency (BCF, Hz) showed no differences among 0.5 mM, 1.0 mM, 2.0 mM BHT and the control groups ($P > 0.05$).

Additionally, post-thaw sperm in samples frozen with 0.5 mM BHT showed extremely significantly higher levels of plasma membrane integrity (PMI, %) than the control samples ($P = 0.001$), whereas, there were no difference among 1.0 mM, 2.0 mM BHT-treated and the control groups (Fig. 1A). Similarly, acrosome integrity (ACRI, %) of post-thaw sperm showed the best in 0.5 mM BHT-replenished group as compared to other groups (Fig. 1B). Supplement of 0.5 mM, 1.0 mM and 2.0 mM BHT to cryomedium can reduce ROS content (%) in post-thaw sperm, respectively; thereinto, only the 0.5 mM treated group

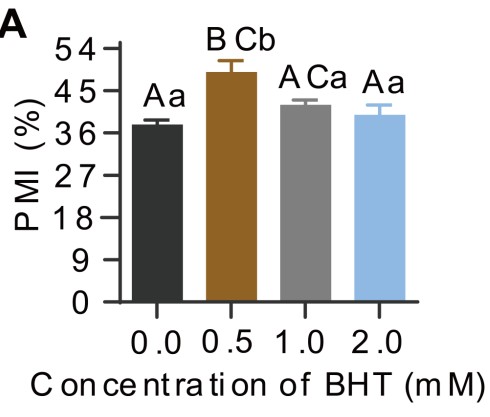

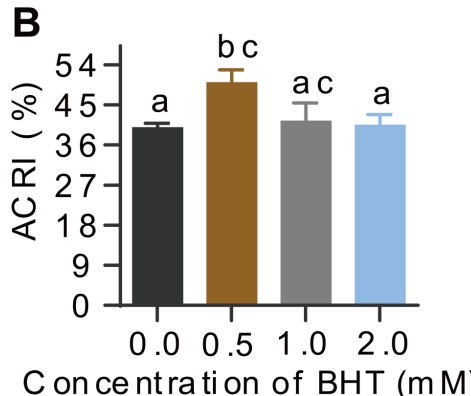

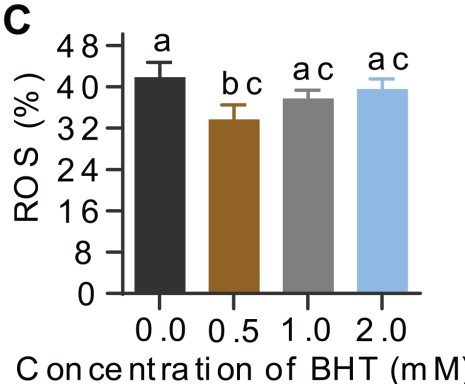

**Figure 1 Effects of BHT on levels of PMI%, ACRI% and ROS of post-thaw goat sperm.** Values with different lowercase superscripts differ significantly ($P < 0.05$) and different uppercase superscripts differ significantly extremely ($P < 0.01$).

presented significantly reduced sperm ROS content ($P = 0.04$) compared to the control group (Fig. 1C). Raw data of this sperm quality-associated indices were shown in Table S2.

## Statistical analysis of MS data

To further explore the underlying molecular effects of 0.5 mM BHT on cryopreserved goat sperm, TMT-based proteomic experiments were performed. Using rigorous statistical filtering, a total of 21,511 unique peptides were identified from 561,255 MS/MS spectrums, with 2,479 corresponding proteins in two groups, of which, 2,476 proteins were quantified (Fig. 2, Table S3).

## Identification of DAPs

According to the screening criteria group ratio (fold change >1.2 or <0.833, $P < 0.05$), fold change of DAPs between the comparable groups (C *vs.* B) was calculated. These DAPs were effectively separated using RStudio (version 3.6.3), as shown in the volcano plot (Fig. 3A), overall, 17 DAPs were identified, where 3 and 14 proteins were highly expressed abundance in C and B groups, respectively (Fig. 3B). All DAPs of goat spermatozoa in C compared to B groups were given in Table 2. Among these, the characterized protein GST class-pi appeared the highest relative up regulation and the 14-3-3 protein theta showed the

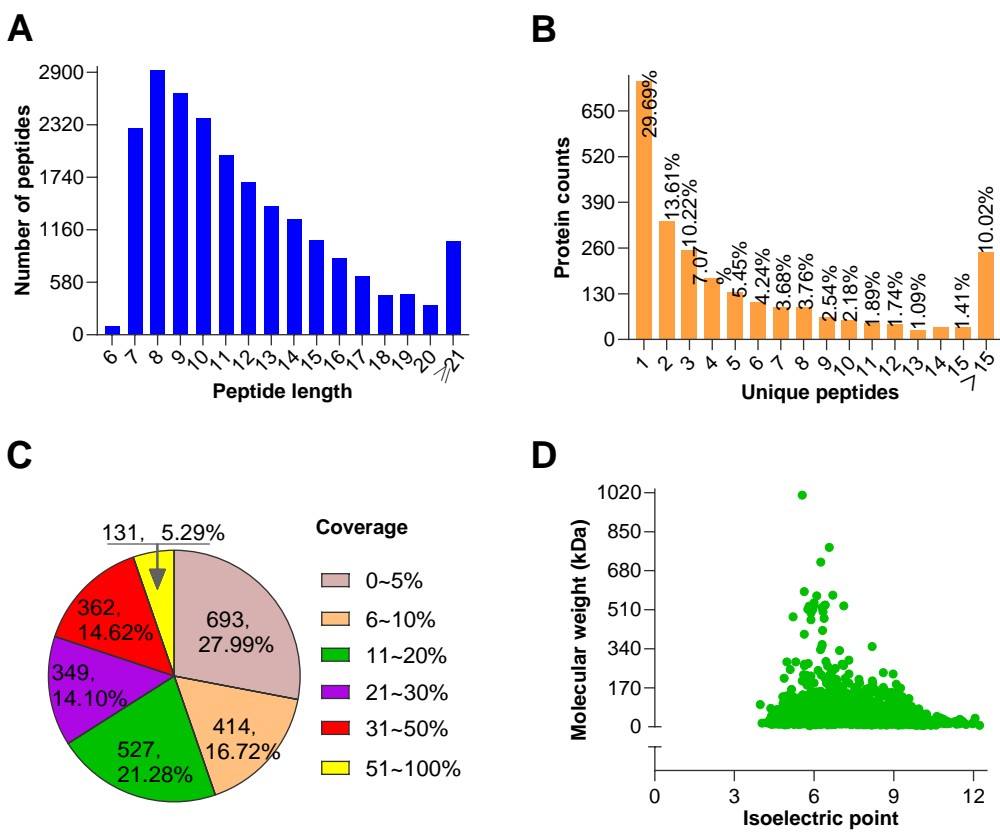

**Figure 2** **Quality control results of proteins.** (A) Distribution of peptide number and length, the abscissa represents the peptide length, and the ordinate represents the number of multiple peptides corresponding to the length. (B) Distribution of unique peptide corresponding to protein, the abscissa represents unique peptide number, and the ordinate represents protein counts corresponding to the unique peptide. (C) Protein coverage (it is defined as a ratio of all unique peptides' length sum to total length of the protein, and is only a reference value related to protein confidence), most proteins coverages were under 30%. (D) Distribution of protein molecular weight and isoelectric point, the abscissa represents isoelectric point of the quantified protein, and the ordinate represents molecular weight of the quantified protein. A large molecular weight range indicates a wide range of quantified proteins.

highest relative down regulation between the comparable groups. These DAPs were mainly distributed in the plasma membrane (23.53%) and unknown subcellular sites (23.53%), followed by cytoplasm (17.65%) and acrosome (11.77%) based on web-server named Euk-mPLoc 2.0 (Fig. 3C).

## PRM quantification

For more authentication of TMT LC-MS/MS proteomic data, we performed PRM verification. Here, the peptide information used to PRM quantification is shown in Table S4, and nine target DAPs with the changed over 1.20 fold and at least two unique peptides for validation, namely, KRT4, KRT5, KRT79, KRT1, ACE, KRT14, SLLP1, KRT3 and IQCF1. Among them, KRT4, KRT5, KRT79, KRT1, ACE, KRT14, KRT3 and IQCF1 were downregulated (C/B ratio < 0.833), whereas, the expression of SLLP1 was upregulated

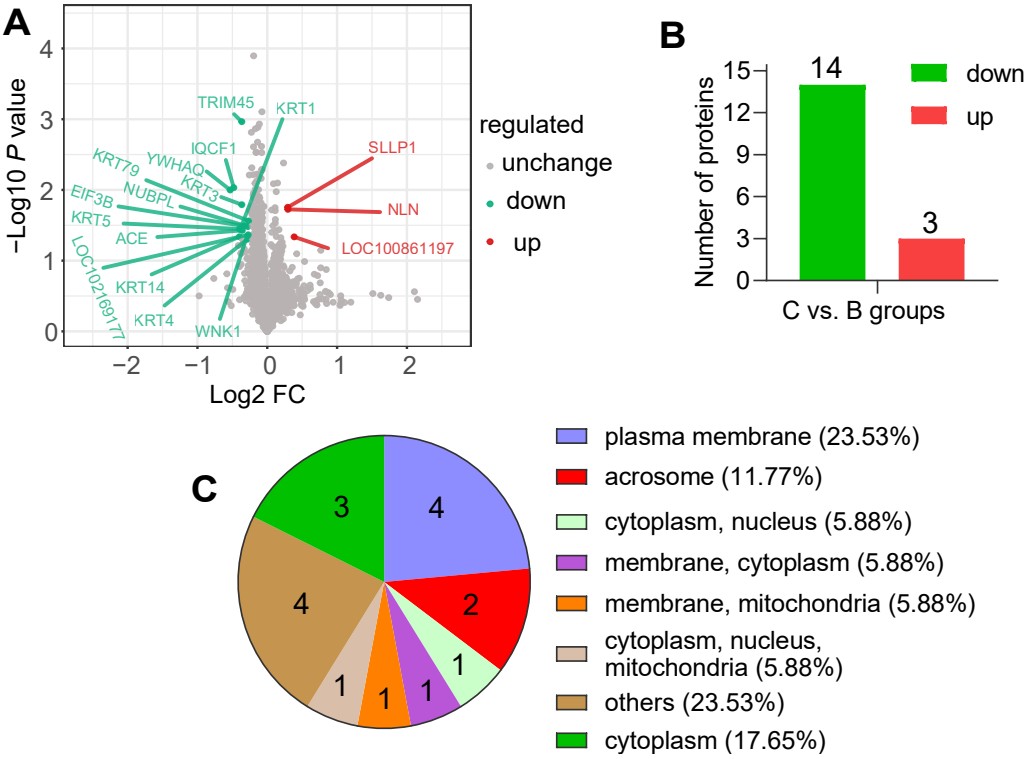

**Figure 3  Screening of the DAPs.** (A) Volcano plots of the comparison (C *vs.* B). The horizontal coordinate represents the fold change of the DAPs (log2). The vertical coordinate represents the P-value (10 is the logarithmic transformation at the bottom). Red points indicate significantly upregulated DAPs; green points indicate significantly downregulated DAPs, and gray points indicate proteins that weren't differential abundance. (B) Counts of DAPs. (C) Subcellular localization of DAPs.

(C/B ratio > 1.20) (Fig. 4). The PRM results showed a similar trend to the TMT results, which indicated that the proteomics data were reliable.

## Functional enrichment analysis based on DAPs

To explore the potential roles of DAPs in the comparable group (C/B), we conducted GO terms and KEGG pathway. DAPs were clustered into 11 GO classes, which contains biological process (BP), cellular component (CC) and molecular function (MF), each DAP was assigned more than one term. Concerning the BP, DAPs were associated with metabolic process, multi-organism process, reproduction, reproductive process, and cellular process; while their enrichment categories included translation regulator activity, structural molecule activity, catalytic activity, and binding in CC terms; and these DAPs were involved in cellular anatomical entity and protein-containing complex in MF terms (Fig. 5A, Table S5). KEGG was utilized for functional pathway annotation of DAPs, there proteins were mapped to the pathways such as renin-angiotensin system and glutathione metabolism pathways (Fig. 5B, Table S6). Additionally, DAPs underwent a protein-protein interaction (PPI) network in the String database (v.11.5, https://cn.string-db.org/). Notably, KRTs family such as KRT1, KRT3, KRT4, KRT5, KRT14 and KRT79 act "cross-talk" nodes

**Table 2  The proteins that were differentially abundant in comparable groups (C vs. B).**

| Accession | Description | Gene name | MW [kDa] | Score | Coverage [%] | Unique eptide | P- value | FC |
|---|---|---|---|---|---|---|---|---|
| A0A452DMF8 | Keratin 5 | KRT5 | 60.2 | 95.13 | 31 | 12 | 0.036 | 0.760 |
| A0A452DT87 | Nucleotide binding protein like | NUBPL | 34.8 | 2.64 | 3 | 1 | 0.032 | 0.801 |
| A0A452DXG9 | Zinc finger protein 474 | LOC102169177 | 52.1 | 2.37 | 2 | 1 | 0.043 | 0.827 |
| A0A452E278 | Eukaryotic translation initiation factor 3 subunit B | EIF3B | 89.3 | 2.50 | 1 | 1 | 0.033 | 0.825 |
| A0A452EAT2 | Angiotensin-converting enzyme | ACE | 150.5 | 159.73 | 17 | 24 | 0.037 | 0.781 |
| A0A452EB32 | IQ motif containing F1 | IQCF1 | 23.5 | 56.42 | 33 | 6 | 0.009 | 0.717 |
| A0A452EBB3 | Non-specific serine/threonine protein kinase | WNK1 | 246 | 2.65 | 0 | 1 | 0.046 | 0.817 |
| A0A452ECX7 | 14-3-3 protein theta | YWHAQ | 27.7 | 32.54 | 23 | 1 | 0.0100 | 0.692 |
| A0A452EJW7 | GST class-pi | LOC100861197 | 23.7 | 8.06 | 15 | 2 | 0.046 | 1.306 |
| A0A452EN33 | Keratin 14 | KRT14 | 55.9 | 67.02 | 23 | 4 | 0.046 | 0.758 |
| A0A452ENV4 | Keratin 79 | KRT79 | 57.8 | 28.53 | 11 | 1 | 0.027 | 0.831 |
| A0A452F5B0 | Tripartite motif containing 45 | TRIM45 | 63.6 | 3.52 | 1 | 1 | 0.001 | 0.777 |
| A0A452FN18 | Keratin 3 | KRT3 | 64.2 | 77.34 | 15 | 3 | 0.016 | 0.777 |
| A0A452FYR1 | Neurolysin | NLN | 80.4 | 3.82 | 3 | 2 | 0.019 | 1.228 |
| A0A452G885 | Keratin 4 | KRT4 | 55.9 | 138.89 | 37 | 13 | 0.049 | 0.820 |
| A0A452GA47 | Cytokeratin-1 | KRT1 | 63.6 | 73.50 | 11 | 4 | 0.036 | 0.765 |
| D7R6C7 | Sperm acrosome membrane-associated protein 3 | SLLP1 | 18 | 49.33 | 31 | 5 | 0.018 | 1.226 |

in the functional modules. They played roles as key intermediate filament proteins that interact with various regulatory proteins, aimed to initiate signaling cascades (Fig. 5C, Table S7).

# DISCUSSION

## Effects of BHT on post-thaw sperm quality are dose dependent

An imbalance between reactive oxygen species (ROS) and antioxidants in favor of the former, this phenomenon results in progression to oxidative stress in cells and tissues (*Banihani, 2017*). In cryopreserved sperm, excessive levels of ROS generated in mitochondria can mediate lipid peroxidation (LPO) in the plasma membrane, coupled with exhaustion of antioxidants, resulting in oxidative injuries, which induce damage of structural and functional components such as proteins, membrane, and DNA in spermatozoa, finally leading to decrease of sperm motility (*Santiani et al., 2014*; *Mislei et al., 2020*), its subsequent ability to approach an egg and successful internal fertilization (*Merino et al., 2015*; *Hyakutake, Sato & Sugita, 2019*; *Merino et al., 2020*; *Park & Pang, 2021*).

Injury to sperm by free radicals generated during the freezing-thawing process could be minimized by supplement of antioxidative protectants in conventional cryo-media, reducing oxidative stress, concurrently, enhancing the motility and membrane integrity while decreased the LPO in sperm (*Thuwanut et al., 2008*; *Gharagozloo & Aitken, 2011*; *Merino et al., 2020*). As a promising antioxidant, BHT has been discussed exclusively in nearly six thousand publications (*Yehye et al., 2015*). Thus far, this lipid-soluble phenolic compound has been confirmed positive effects on cryopreserved sperm (*Jara et al., 2019*; *De*

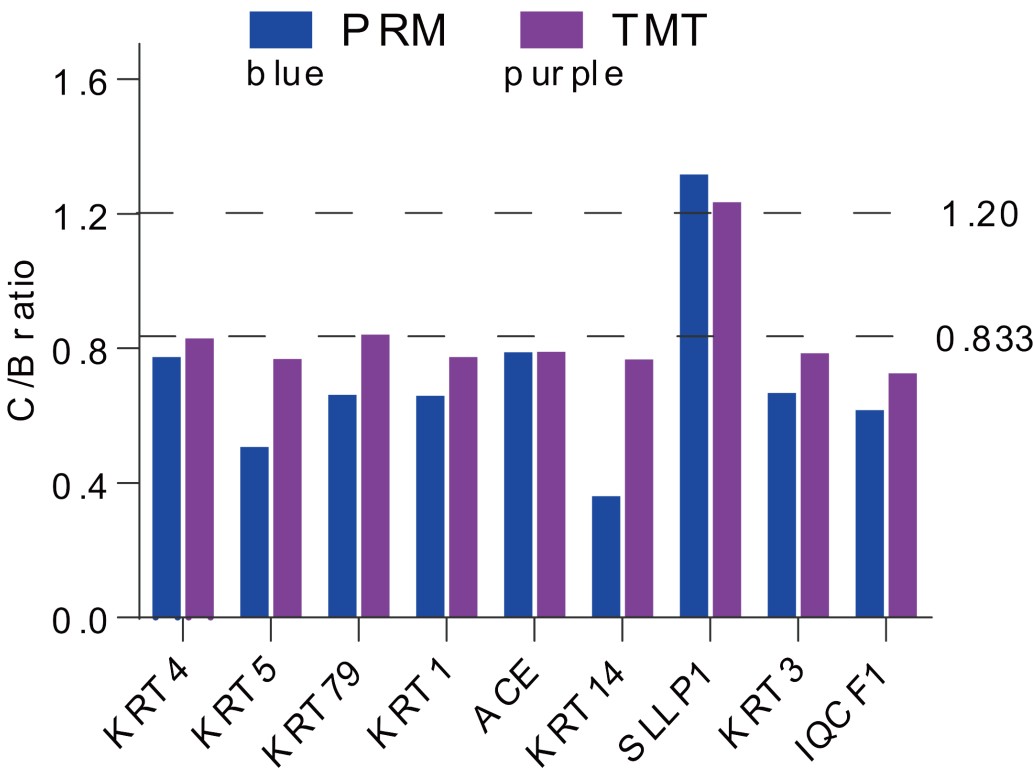

**Figure 4 Expression patterns of selected DAPs using TMT analysis and PRM validation.** From top to bottom, two dotted lines represent 1.20-fold (up-regulation) and 0.833-fold (down-regulation), respectively.

*Andrade et al., 2023*). Specifically, BHT readily penetrate the sperm membrane, enhancing its fluidity and influencing the membrane phase transition. This action potentially prevents ice-crystal formation within sperm cell (*Hammerstedt et al., 1976*; *Khumran et al., 2015*). Additionally, penetrated BHT also acts primarily as a proton donor for the free radicals and the regenerate acylglycerol molecule, or it can reduce sites suitable for molecular oxygen attack, and terminate oxidation of the free-radical chain reaction, thereby decreasing the harmful effects of ROS on sperm during freezing/thawing process (*Fujisawa, Kadoma & Yokoe, 2004*; *Osipova et al., 2016*; *Fasihnia et al., 2020*). There are available references on the effects of BHT added to semen freezing extenders to protect sperm during cryopreservation in a dose dependent manner. For example, addition of 0.5 mM and 1.0 mM BHT appears to be optimal for the cryopreservation of human semen, due to its antioxidant property for improving the progressive sperm motility and reducing ROS production compared to the control (*Ghorbani et al., 2015*; *Merino et al., 2015*). In canine, supplementation of 0.2–0.8 mM BHT in the cryo-media did not affect the cryopreserved sperm motility, viability and acrosome integrity whilst 1 mM or 1.5 mM BHT significantly improves sperm plasma membrane (*Neagu et al., 2010*; *Sun et al., 2020*). Additionally, higher values of the sperm motility, average path velocity, GPx activity, and acrosome integrity in the cryo-medium supplement of 1.5 mM BHT than those in the control (*Neagu et al., 2010*). However, these

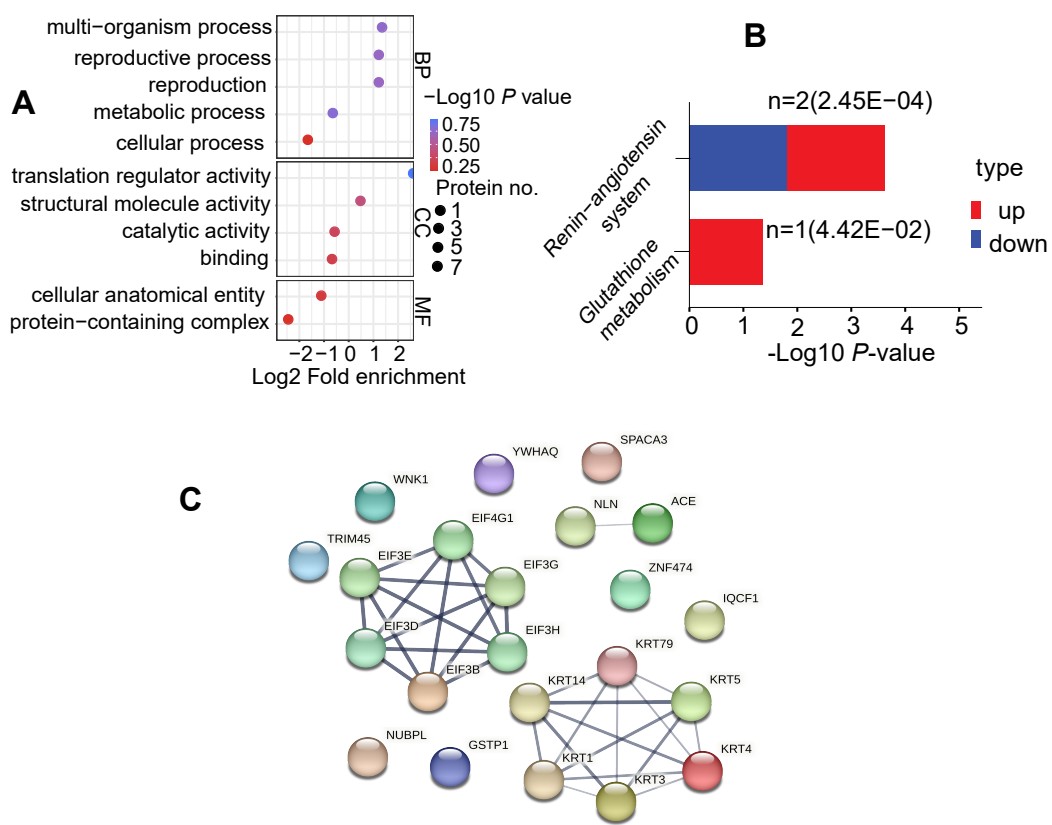

**Figure 5  Functional enrichment analysis of DAPs.** (A) GO analysis for DAPs. Three functional domains were displayed that including biological process, cellular component and molecular function terms. (B) Analysis of KEGG pathways in the comparable group (C *vs*. B). (C) Protein-protein interaction network diagram of DAPs

characteristics of chilled sperm reversed when BHT concentration reached 1.6 mM in the extender (*Sahashi et al., 2011*). In Murrah buffalo bull, 0.5–1.0 mM BHT-supplementation in freezing extenders significantly increased the progressive motility, viability, and acrosome integrity of frozen thawed spermatozoa compared to the control (*Nain et al., 2022*). Expansively, 0.5–2.0 mM BHT-supplementation in extenders significantly decreased lipid peroxidation of cryopreserved boar sperm in relation to the control, thereinto, motility, membrane and acrosome integrities, fertilizing ability of post-thaw sperm were the highest by addition of 1.0 mM BHT (*Trzcinska et al., 2015*). Nevertheless, our study showed that 0.5 mM BHT-supplementation in semen extender was the optimal concentration for improving the motility (TM, PM), plasma membrane and acrosome integrities, and decreasing levels of ROS in cryopreserved goat sperm compared to the control, which effectively improved the quality of frozen sperm.

## BHT modifies the protein profile of goat sperm during cryopreservation

The TMT based quantitative proteomic technique was applied to evaluate effects of BHT on the proteome of cryopreserved spermatozoa in Yunshang black goats. After bioinformatics analysis, overall, 17 DAPs involved in sperm characteristics and functions were screened between C and B groups. Those proteins were mainly involved in sperm-egg binding and fertilization, RNA transport, estrogen signaling, structural molecule activity and glutathione metabolism *etc.*, which may be associated with the decline of sperm quality after cryopreservation, and the molecular role of BHT in reversing this adverse state.

For instance, sperm acrosome membrane-associated protein 3 (SLLP1) and GST class-pi (LOC100861197) were more abundant in C group. SLLP1 (also referred to as SPRASA) has been identified firstly in the acrosome of human sperm and involved in immune-mediated infertility (*Chiu et al., 2004*). As a member of c-type lysozyme/alpha-lactalbumin family, SPRASA has an exon-intron organization and sequence conservation, similar to c-type lysozymes (*Chiu et al., 2004*; *Wagner et al., 2015*). Afterwards, it has been reported that the protein could be a target for anti-sperm antibodies in some infertile male, playing possible roles in sperm-egg bonding process, as well as subsequent development of early embryo in hamster, murine or bovine models (*Prendergast et al., 2014*). In this study, the expression of SPRASA was higher in cryopreserved goat spermatozoa without antioxidant cryo-protection, suggesting that this protein may be a potential infertile marker of frozen-thawed goat sperm, and its specific molecular function is worth further exploration. Glutathione-S-transferases (GSTs) have been demonstrated to be present on the goat sperm surface that serve as zinc-responsive antioxidants to bind oocyte (*Aravinda et al., 1995*; *Hemachand & Shaha, 2003*; *Chung, Walker & Hogstrand, 2006*). Remarkably, the isoform of GSTs, namely GST-Pi has recently been shown to be present primarily in sperm plasma membrane and is responsible for binding to the zona pellucida (*Kumar, Singh & Atreja, 2014*). Previous report has been shown that GST-Pi expression in relation to oxidative stress and GST activity (*Huang et al., 2004*). In goat cryopreserved sperm, GST-Pi was up-regulated and the higher levels of oxidative stress, which suggesting that GST-pi expression in sperm with higher levels of oxidative stress may not be enough to eliminate the harmful effects of ROS.

On the contrary, keratins (KRT1, KRT3, KRT4, KRT5, KRT14, KRT79), IQ motif containing F1 (IQCF1), nucleotide-binding protein like (NUBPL), and angiotensin-converting enzyme (ACE) were more abundant in B group than in C group. Interestingly, Keratins are typical intermediate filament proteins, play roles in protecting cell/tissue from stress, and act as biomarkers for some organ diseases (*Moll, Divo & Langbein, 2008*; *Mun, Hur & Ku, 2022*). Here, we identified up-regulated keratins proteins KRT1, KRT3, KRT4, KRT5, KRT14 and KRT79 in cryopreserved goat sperm which treated with a freezing medium containing BHT antioxidant. These keratins as rope-like structures may be involved in microtubules or tension-bearing role in sperm flagella to maintain sperm motility (*Kierszenbaum, 2002*; *Hinsch et al., 2003*).

As a novel acrosomal protein, IQCF1 is proved to interact with calmodulin on the sperm head and functioned in sperm motility, additionally, this protein is associated with sperm

capacitation, especially sperm protein tyrosine phosphorylation and the membrane fusion events during acrosome reaction (*Bendahmane, Lynch & Tulsiani, 2001*; *Fang et al., 2015*). Although it is tempting to hypothesize that IQCF1 expression is correlated with sperm capacitation, more studies are warranted to confirm this finding.

NUBPL is one of the essential subunits for sperm mitochondrial complex I (MCI) assembly, typically, activity of MCI effectively maintains the optimal levels of ROS (*Chai et al., 2017*; *Cheng et al., 2022*), we thereby concluded the indirect effect of NUBPL on ROS production.

For ACE, which has two isoforms, thereinto, the testicular isoform of ACE (tACE) is expressed in haploid elongating spermatids and sperm. Of note, tACE plays an important role in sperm fertilization because of its dual activities of dipeptidase and a GPI-anchored protein releasing factor, and correct positioning and distribution in the sperm membrane is prerequisite for the fertility (*Sibony, Segretain & Gasc, 1994*; *Deguchi et al., 2007*; *Pencheva et al., 2021*). *Ojaghi et al. (2018)* suggested that freezing and thawing process could reduce the abundance level and activity of tACE in bull sperm. Our finding of tACE has a significant increase in the expression level in highly motile goat cryopreserved sperm which being storage in cryomedium contains BHT. Therefore, we may conclude that tACE indeed associates with sperm fertilization competence, it could serve as marker for fertilizing ability of spermatozoa.

## CONCLUSIONS

The utilization of antioxidants during cryopreservation has emerged as a promising approach to mitigate detrimental effects of ROS on sperm quality. Our study has underscored the significant enhancement in goat sperm quality parameters after freezing/thawing, when the extender was supplemented with 0.5 mM of BHT. Furthermore, the over-expression of certain proteins, such as SLLP1, GST-Pi, IQCF1, NUBPL and tACE were observed, suggesting their potential as novel biomarkers for appraising post-thaw sperm quality and fertility in the goat. As the field advances, a deeper understanding of these proteins and their interaction with antioxidants like BHT will be crucial for refining cryopreservation protocols and enhancing the success rates of AI in goats.

### Funding

This research was supported by the Major Science and Technology Project of Yunnan Province (Grant/Award Number: 202102AE090039), the Yunnan Applied Basic Research Projects (Grant No. 202301AS070005), and the Earmarked Fund for China Agricul-ture Research System (Grant No. CARS-39-08). The funders had no role in study design, data collection and analysis, decision to publish, or preparation of the manuscript.

### Grant Disclosures

The following grant information was disclosed by the authors:

Major Science and Technology Project of Yunnan Province: 202102AE090039.
Yunnan Applied Basic Research Projects: 202301AS070005.
The Earmarked Fund for China Agricul-ture Research System: CARS-39-08.

## Competing Interests

The authors declare there are no competing interests.

## Author Contributions

- Chunyan Li conceived and designed the experiments, performed the experiments, analyzed the data, prepared figures and/or tables, authored or reviewed drafts of the article, conceptualized the study, and approved the final draft.
- Larbi Allai performed the experiments, analyzed the data, prepared figures and/or tables, authored or reviewed drafts of the article, and approved the final draft.
- Jiachong Liang performed the experiments, prepared figures and/or tables, validated the experiments, and approved the final draft.
- Chunrong Lv performed the experiments, prepared figures and/or tables, validated the experiments, and approved the final draft.
- Xiaoqi Zhao performed the experiments, prepared figures and/or tables, and approved the final draft.
- Xiaojun Ni performed the experiments, prepared figures and/or tables, and approved the final draft.
- Guoquan Wu performed the experiments, prepared figures and/or tables, authored or reviewed drafts of the article, investigated resources, and approved the final draft.
- Weidong Deng performed the experiments, prepared figures and/or tables, analyzed formalization, and approved the final draft.
- Bouabid Badaoui analyzed the data, authored or reviewed drafts of the article, and approved the final draft.
- Guobo Quan analyzed the data, authored or reviewed drafts of the article, and approved the final draft.

## Data Availability

   The raw data is available at the iProX partner repository: IPX0006546000.

## Supplemental Information

Supplemental information for this article can be found online at http://dx.doi.org/10.7717/peerj.17580#supplemental-information.

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
