# Peer review of "The antioxidant effects of butylated hydroxytoluene on cryopreserved goat sperm from a proteomic perspective"

_PeerJ, doi:10.7717/peerj.17580_

## Round 0.1 · original submission · Major Revisions

Dear authors, I ask you to carefully analyze the reviewers’ comments and format the manuscript in accordance with the requirements of the journal. Please also consider these technical notes.

All equipment and computer programs must be accompanied in brackets with information about the manufacturer (software developer), city, country and year.
Sources in the same brackets in the text of the article should be arranged in year order.

The discussion needs to be structured into subsections. This will increase the clarity of the presentation of the material.

Journal titles must be formatted in accordance with journal design standards.

Punctuation marks in bibliography are often placed randomly.

**Language Note:** The review process has identified that the English language must be improved. PeerJ can provide language editing services - please contact us at [email protected] for pricing (be sure to provide your manuscript number and title). Alternatively, you should make your own arrangements to improve the language quality and provide details in your response letter. – PeerJ Staff

·

Basic reporting

This is an interesting and useful investigation of the proteomic impacts effects of BHT supplementation for goat sperm cryopreservation. This is an emerging field of study, and so consideration of antioxidant effects is useful in driving the field forward. The work is well researched and well supported with relevant citations throughout. The figures and tables are generally well formatted (though inclusion of the full terms for variables, rather than just their acronyms, would be useful here). The raw data are shared within an excel file and the results address the stated rationale.
The main area to develop is the grammar for this manuscript. There are numerous grammatical errors in the work, including changes of tense, and this makes the communication of the underlying science more challenging. I have added specific points on the PDF of this manuscript, though I would recommend a full proof read from the authors.

Experimental design

The work fits well within the scope of PeerJ and the rationale for the study is clearly stated. The novelty of the work is clearly shown, though there could be more elaboration as to the future directions for this work. The methods are generally well described, though there is reference to pooling of sperm samples. Is this pooled per animal, or pooled cross animal? This should be discussed in a bit more depth. Similarly, include information on ages of animal, for example. This said, the majority of the methods are clear and repeatable.

Validity of the findings

The data are provided and replication should be possible based on the amount of information provided. The conclusions could be made a little clearer. The test statistics should be provided alongside the p values in order to repeatability.

Additional comments

This is an useful study with some potentially exciting consequences in terms of cryopreservation. As such, there is some scope for publication. My main areas to address are as follows:
1. Check repeatability of the methods and clarify on the pooling of samples (as there may be differences between individuals in terms of sperm parameters).
2. Include relevant test statistics when quoting P values.
3. Wording. Conduct a full proof read to correct grammatical errors.

·

Basic reporting

Data on bioethical expertise should be added to the manuscript.

Experimental design

Study limitation should be added to the manuscript

Validity of the findings

The possible explanation is required why only 0.5 mM BTH significantly reduces sperm ROS content. 2. Most of antioxidants are present not in sperm cells, but in seminal plasma. Information about this should be added to manuscript

---

## Round 0.2 · accepted · Accept

Your article is recommended for publication. I hope that the results of your experiments will help improve technology in this field of medicine and veterinary medicine.

·

Basic reporting

Thank you for providing a revised version of your manuscript. The edits to the work are clear and my initial concerns regarding the clarity of the methods have been well addressed. Raw data are provided as a supplementary file. The work is now in a stronger position overall.

Experimental design

I initially raised concerns regarding pooling of samples, as this was not clear in the original manuscript. These points have now been clearly addressed and the work is more repeatable as a result. As previously stated, this is an useful study that fills a gap in the cryoconservation literature, yet it has clear links to the existing research.

Validity of the findings

The findings are now much clearer. Substantial revisions have been made to the introduction, methods and discussion, and so the conclusions are now more clearly stated.

Additional comments

Thank you for providing the revisions to your work. This study is now much clearer overall.

·

Basic reporting

Previous remarks were taken into account and corrected.

Experimental design

There are some limitations in the present study. First of all a limited number of animals from which samples were obtained. This should be added to the discussion.

Validity of the findings

Previous remarks were taken into account and corrected.